# Network Governance and the Evolving Urban Regeneration Policymaking in China: A Case Study of Insurgent Practices in Enninglu Redevelopment Project

**Nannan Zhao** [1,2]  **, Yuting Liu** [1,]***  and June Wang** [2]

1   Department of Urban and Rural Planning, South China University of Technology, Guangzhou 510640, China; nannazhao3-c@my.cityu.edu.hk or nn.zhao@foxmail.com
2   Department of Public Policy, City University of Hong Kong, Kowloon 999077, Hong Kong; june.wang@cityu.edu.hk
*   Correspondence: ytliu@scut.edu.cn

**Abstract:** The network governance approach has been adopted by many researchers and practitioners with respect to policy analysis and modern state governance. This study utilizes a broadly defined network-based framework to trace the evolution of urban regeneration policymaking in Guangzhou, China. Drawing upon the notions of "network" and previous scholars' work on participatory planning, this study focuses on the changing relational networks among the various actors that are engaged in the urban regeneration process and the factors motivating these changes. In so doing, this study uses the ongoing Enninglu redevelopment project (2006–) as an illustrative case study. By examining the insurgent practices in the Enninglu redevelopment process, this study argues that urban redevelopment policymaking in China has changed twofold. First, the planning regime has transited from state-dominant practices to one that is primarily driven by the local government, the enhanced role of higher education institutions and experts as a "professional interest group", and the increased participation of non-state actors in the policymaking process. Second, the decision-making mechanism has transformed from an interventionism-oriented system to a polyarchy-oriented system in which both the advocacy coalition and opposition coalition are embedded in the governance network. Additionally, the emergence of insurgent practices in Enninglu suggests an emerging shift toward substantive participatory governance in the Chinese context. From a network perspective, this study attempts to contribute to the understanding of the evolving urban regeneration policymaking in China and broader governance networks in urban regeneration practices.

**Keywords:** urban regeneration; network governance; planning regime; insurgent practices; participatory planning

## 1. Introduction

Discussions on Western democracy have long dominated the direction of urban studies. The resulting debates have merged into governance literature in light of manifested state failures or market failures [1]. Many pluralists, such as Lindblom and Dahl [2], regarded a pluralistic social order, a competitive market economy, and public participation as three necessary conditions for achieving a polyarchal or democratic regime, which can contribute to "good" governance. In the planning sphere, similar arguments on elite democracy and political inequality are reflected in some counter-hegemonic discourses and actions. Theoretical debates have erupted into conflicts among some planning scholars who advocate Habermas's [3] communicative action theory, such as Healey [4] and Forester [5], thus clashing with political "realists" who advocate interventionist strategies. As Holston [6] mentioned, modernist planning both relies on and is built up by the state and the insurgent practices. It characterizes the guiding principles for insurgent planning practices, such as counter-hegemonic, transgressive, imaginative, and spatialized

practices [7,8]. Therefore, in light of the Foucauldian notion of governmentality, the key question is not about whether the government or elite-based interest groups should or should not intervene in urban planning activities, but is about how to build and manage effective networks so that urban planning can better serve the public interest.

To address this question, it is essential to fully understand the evolving relational networks of various actors and their roles in the policymaking process. In the facet of urban questions, many scholars have adopted theories of "urban regime" or "growth coalition" to analyze the public–private partnership and local politics during urban (re)development [9,10]. However, given the particular focus on the economic aspect, the classic urban regime theory has obvious limitations in terms of explanatory power, and fails to take into account civil society and all kinds of other interest groups that wish to accumulate social or political capital rather than merely economic capital [11–13]. As a response, the 1990s saw explosive developments in the "network" theory in which non-state actors and civil society are highlighted [14–17]. In this sense, the idea of "network governance" focuses on the horizontal, decentralized, and interactive relations between independent (but interdependent) actors that share a high degree of trust within inclusive participatory decision-making activities [18–21]. On this basis, this study argues that "network governance" or "governance-beyond-the-state" can provide fruitful analysis in terms of the extent to which diverse actors coexist within a project-oriented network, and their capacity to act collaboratively in urban regeneration policymaking.

By tracing the evolution of urban regeneration policymaking in China, this study argues that the actors, policies, discourses, and actions have undergone profound changes. Many previous scholars have provided in-depth analyses and critical views of the state–society relations and power relations in China's urban regeneration policymaking under the interwoven nature of capitalist globalization and neoliberalization [22–25]. Nevertheless, empirical research on how non-state actors and civil society form stakeholder-based relational networks through formal or informal approaches is still lacking [26]. Moreover, it is uncertain whether this new form of governance has yet been institutionalized into the Chinese planning regime. Therefore, this study first attempts to answer how urban regeneration policymaking has evolved in China, and whether network governance is useful for understanding this process on the basis of the broadly defined notion of "network governance". Then, the remainder focuses on the insurgent practices in the Enninglu redevelopment to analyze how networks and coalitions have taken shape both organically and informally, and how advocacy coalitions and opposition coalitions have both utilized participation as an efficient instrument for legitimization.

## 2. Institutionalization of Network Governance in the Planning Regime

### 2.1. "Network" Theories and an Analytical Framework

In complex modern societies, urban (re)development policymaking is regarded as a socially constructed process in which multi-scalar informal actors are involved [16,27]. In China, the government plays the most powerful role in terms of resource integration and (re)distribution, public welfare and service production, and social cohesion and organization. However, with today's globalization and neoliberalization, the Chinese government has had to confront and adapt to the pluralist trends in the social, economic, and political domains [22,28,29]. In this regard, "network" studies are developed to better understand the unique challenges of cities operating in a context where hierarchal bureaucracy no longer provides the primary tool for socio-spatial (re)configuration [15,30]. Healey [15] summarized three "logics" in structuring urban processes: the logics of market, hierarchy, and networks. Instead of nesting in a hierarchical model of levels of governmental responsibility, this new arena of urban governance is characterized by non-hierarchic and consensus-seeking decision-making procedures, which has attracted a variety of different groups of actors from CPPCC (Chinese People's Political Consultative Conference) members to local community organizations and residents. In this regard, the advantages of a network perspective in urban governance studies are twofold.

First, the importance of relational networks of multi-scalar actors has been highlighted under the participatory planning thesis [31,32]. Despite decades of debates, participatory and communicative planning continues to be contested. Nevertheless, the community-based participatory approach has been adopted by many researchers and practitioners as a preferred approach in addressing complex place-based governance issues and socio-spatial conflicts [33–36]. However, in light of the limitations of ideal participatory governance, Friedmann and Castells further discussed the possibility of radical planning, in which civil society or so-called insurgent actions play a vital role. In this regard, the notion of insurgent actions as radical and participatory planning practices responds to the government-backed planning regime through inclusive governance [7,36]. More recently, research has documented an emerging transformation in the planning regime in which participation works as a post-political tool—a means to depoliticize planning and to legitimize neoliberal policymaking [37]. Unlike Western countries, insurgent practices in the Chinese context may face more difficulties. This study seeks to understand how civil society actors form a discourse coalition and take action to challenge institutional planning interventions.

Second, the idea of "network governance" or "governance-beyond-the-state" can fulfill the gap of existing political–economic logic in analyzing the socially and politically constructed urban process. Network governance refers to collaborative decision-making by complicated self-organized and inter-organizational networks, including public agencies and non-state agencies (who are normally marginalized groups) [16,38,39]. This is essentially different from the logic of the market or hierarchy. Governance in this new form can be accomplished either formally, for instance, through regular meetings of selected organizational representatives, or more informally through ongoing (but typically insurgent) practices adopted by people who have a stake in the network's success [7,40]. In the planning sphere, government-led or outcome-oriented urban (re)development arrangements have proved to be contentious, costly, and inefficient, with strong side-effects concerning the inherited socio-spatial relations such as displacement or gentrification.

This paper argues that the performance of participatory planning in urban regeneration policymaking ultimately depends on whether decision-making power permeates from the core to the periphery of the network, as shown in Figure 1. When narrowing this down to examine the area-based governance arrangements that involve multi-scalar interactions, the analytical framework in Figure 1 can be useful. Based on the theories of Healey [15] and Coaffee and Healey [41], this framework contains three levels of power. The first level is specific episodes in which interactive practices take place in different *action arenas*. At this level, the most important question is to identify the key actors and their action arenas. In this study, the initiatives at the episode level refer to the insurgent actions taken by local community oppositions. The second level is the governance process, which focuses on the mutual relationship of these actors and their performances in terms of insurgent initiatives. In this study, the evolution and formation of governance networks and coalitions will be analyzed in terms of stakeholder-based practices and discourses. The third level is the cultural assumptions held by different social groups about urban agendas or governance practices. This level focuses on the legitimation and impact of the episodes and governance processes. This is the reformulation and reconfiguration of inherited structures, such as the authority of rules, the justice of resource allocations, and the inclusiveness of accountability. In this regard, this study argues that the formulized networks and coalitions are based on a high degree of trust and resonance with cultural assumptions.

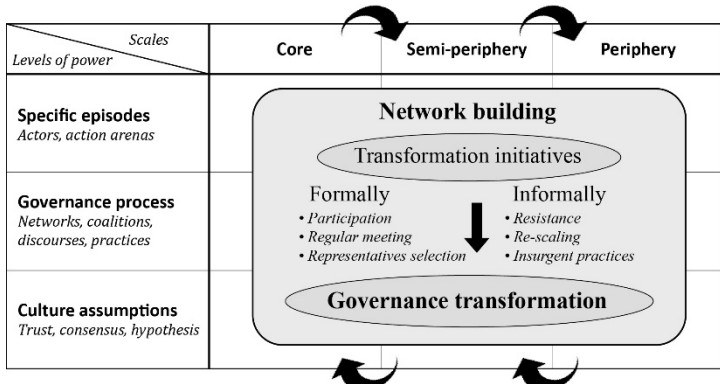

**Figure 1.** An analytical framework of network governance in China's urban regeneration policymaking (modified from Healey, 2006a).

*2.2. Network Governance in the Urban Regeneration Regime*

Network governance provides an empirical perspective beyond the public–private partnership in order to analyze collaborative decision-making—whether in an active or passive manner—in urban regeneration. On one hand, the experience of Western countries indicated a governance transformation in the urban regeneration regime. In the aftermath of World War II, urban renewal and regeneration, as a policy-oriented act, sparked widespread debates about the relationships between the state and all kinds of power elites and interest groups, the government and citizens, and the central government and the multi-scalar local government [42–44]. As early as the 1960s, Jacobs [45] strongly criticized the highly centralized power relations in the large-scale urban redevelopment movement in the United States. Over the past two decades, the focus of urban regeneration in Western countries has gradually changed from being oriented by policy goals, which primarily involved property-led redevelopment that was dominated by either the public or private sectors, to a broader mix of socio-spatial networks and a far greater emphasis on the interactive process among diverse actors and the ideal partnership in community-based policymaking [42,46,47].

On the other hand, the evolution of the urban regeneration regime in contemporary China has shown a similar trend toward community-based participatory planning, but with a very different governance focus and process. Since the 1980s, China has gone through rapid economic development, and for a long time, economic growth dominated every aspect of urban development. In this regard, the Chinese Communist Party (CCP) and the central government has played a vital role in promoting China into the global network. However, since the 2000s, radical changes in the social, economic, and political spheres have forced a change in the role of the government. Some obvious changes refer to capitalist globalization and neoliberalization, decentralization and fragmentation, computerization and informatization, and so on. These emerging trends in modern societies resulted in the gradual blurring of the frayed boundaries between the public and private sectors, the state and society, and the socialist social order and market-led capitalist social order. For instance, in recent years, the redevelopment agenda in the inner-city areas in Guangzhou has faced sharp challenges due to demolition–conservation controversies and social resistance. This study thus argues that the evolving urban regeneration in China is a contentious, multi-scalar, and network-oriented process. The following sections focus on the evolution of the urban regeneration policymaking in Guangzhou, China. Then, based on the aforementioned network-based framework, the case study of the Enninglu redevelopment project is used to examine the actors and their power relations, and to analyze the impacts of these insurgent practices and the potential cultural assumptions of networks and coalitions.

## 3. Methodology and Data Collection

### 3.1. Study Area and Research Scope

Enninglu is an inner-city neighborhood located in Liwan District, as shown in, that has long been the urban commercial and residential center of Guangzhou City. In 2006, the Guangzhou Municipal Government proposed several urban redevelopment schemes to upgrade its urban image for the 2010 Asian Games; the Enninglu regeneration project was one of these programs. In May 2007, in conformance with the vision at the municipal level; that is, the strategy of "revalorizing the central city area", Liwan District published its urban renewal plan and proceeded with the demolition of 11.37 hectares (of land area) in the Enninglu neighborhood, as shown in Figure 2. The project involved 1352 dwellings and 1965 households (including 1248 private housing ownerships) with a total floor area of 207,134 m$^2$. The present study draws on an investigation of the regeneration processes and policymaking in Enninglu from 2006 to 2020, and presents evidence of the ways that urban redevelopment objectives and projects are established, contested, and developed. As the first pilot regeneration project after the market first engaged in urban redevelopment in Guangzhou, the ongoing project in Enninglu can be used as a salient example for analyzing the networked relations between diverse actors.

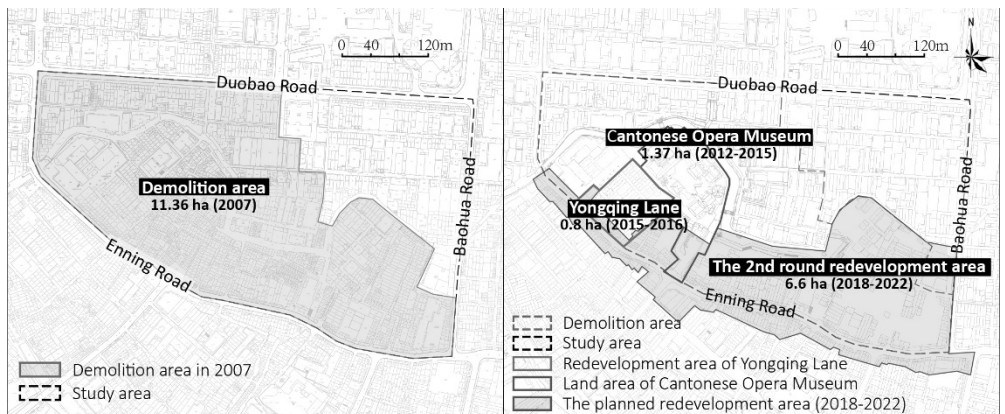

**Figure 2.** The first demolition plan of Enninglu in 2007 and the newly established redevelopment projects.

Unlike many other urban regeneration projects, Enninglu is located in a historic urban central area, and therefore the redevelopment practices have had more impact on socio-economic, cultural, historical, and landscape contexts. Without a robust collaborative partnership like that in Liede Village [48], the local authorities in Enninglu have had to deal with individual residents and many other stakeholders, such as environmentalists, local media, radical scholars, and local focus groups. In this neighborhood, more than 60% of the existing dwellings are privately owned by individuals, and 17% are of mixed ownership. It turns out that the inappropriate and radical demolition of these private houses and mixed-property houses has led to strong social resistance and continuous local initiatives in the following regeneration [49]. In response to these redevelopment concerns, the Bureau of Land Resources and Housing Management of Guangzhou Municipality has had to postpone the demolition. In the interval of the demolition postponements, several public-invested projects have been initiated, such as the Cantonese Opera Museum and the remediation project of the Enning Canal landscape, as shown in Figure 2. Since August 2018, a new stage of the urban redevelopment project, launched by the Guangzhou Liwan District Government, has been constructed on the demolished parts of Enninglu, as shown in Figure 2. Part of the project was completed prior to September 2019 for the sake of the political performance on National Day. During this process, despite it being argued that participation has been adopted by the authorities as a post-political tool as a means to depoliticize planning and legitimize neoliberal policymaking, the network of actors has changed. Instead of arguing whether the outcome is positive or not, the present study

focuses on the governance processes during the evolving urban regeneration policymaking, in which complicated networks and coalitions have taken shape.

*3.2. Data Collection*

This study is based on the continuous in-depth fieldwork in Enninglu between 2017 and 2020. As a practitioner and observer in the Enninglu regeneration project, the author collected a range of first-hand dynamic data in urban redevelopment practices. During the fieldwork, this study primarily employed the interview method to obtain qualitative data. First, in-depth interviews were conducted with key actors of the past two years including local authorities (e.g., the Urban Renewal Authority of Guangzhou and the Liwan District Government), street-level bureaucracy officials and organizations (e.g., the Sub-District Office and Neighborhood Committee), real estate developers (i.e., Vanke), planners and experts, and more than 20 local community residents. Specifically, this research adopted the participatory action research (PAR) approach to get involved in residents' daily life. The author was invited by local residents to enter their homes, investigate their housing conditions, and conduct in-door interviews, as shown in Figure 3. Second, a participatory focus group approach was adopted to examine the mutual relationship among the diverse actors. For instance, during 2018 and 2019, the author participated in all seven meetings held by the Enninglu Co-Creation Committee, as shown in Figure 4. In this process, the author collected first-hand data about how different actors bargained and negotiated with each other. The networks and collations gradually took shape during this process either formally, for instance, through regular meetings or negotiations of selected representatives, or more informally through ongoing but typically insurgent practices and resistances. Last, observational works were also conducted during field investigations, public participation, and community negotiations, in order to capture the changes of both physical environments and relational social network. Therefore, this study is essentially an institution and policy analysis that relies on a combination of documentary analysis, in-depth interviews, and participatory observations.

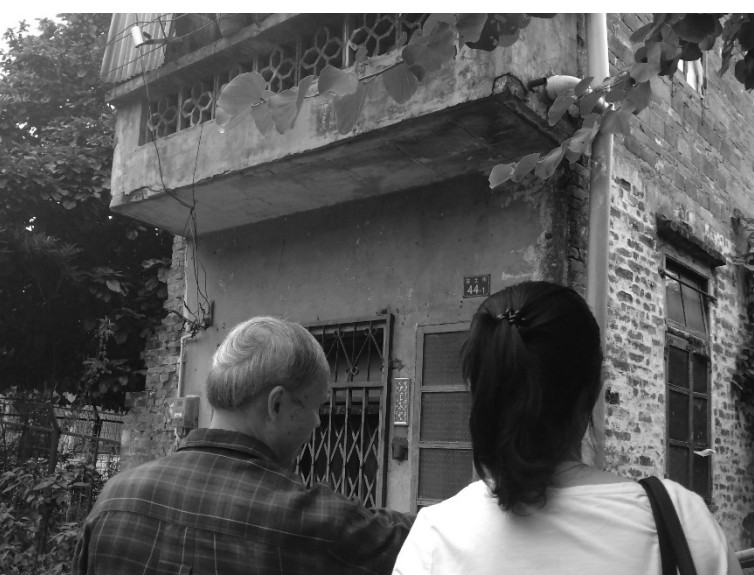

**Figure 3.** Photo of field investigation in Enninglu.

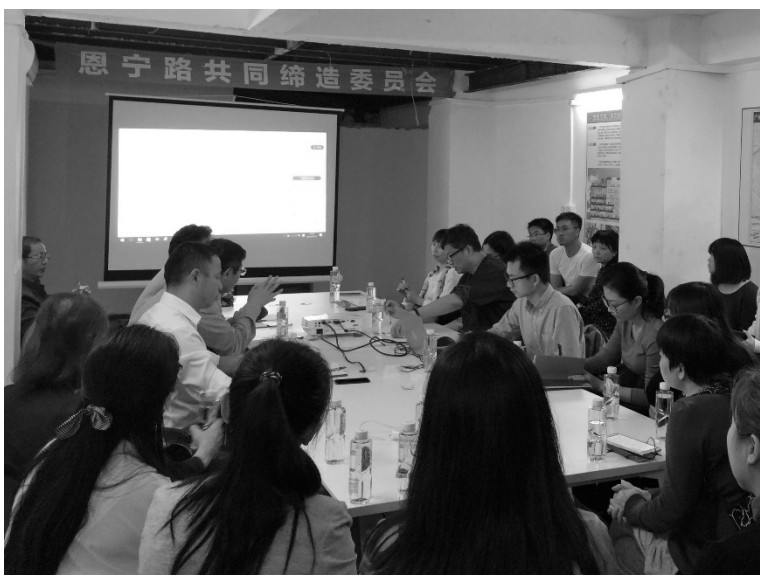

**Figure 4.** Photo of negotiation meeting with the government, developer, and local resident.

## 4. The Network-Based Governance Dynamics in the Evolving Urban Regeneration Policymaking in Guangzhou

### 4.1. Growth-Oriented Urban Expansion Strategy (the 1980s): From Party-Directed to Central-Government-Driven

Between 1949 and 1978, the CCP dominated the public sphere in urban economic development and spatial construction [50] by controlling the distribution of access to dwellings and welfare benefits. Thus, the CCP is at the very core of urban development policymaking, as shown in Figure 5. Since the 1978 reform and opening-up was initiated, Deng Xiaoping proposed a new ideology—the "Separation of Party and Government"—which aimed to distinguish the different functions, responsibilities, and tasks of the CCP and the government agencies. During this period, although the power of final decisions and scrutiny was still held by the CCP, the decisive power gradually relinquished to the government bureaucracy and other vested interest groups, as shown in Figure 5. As shown in Figure 5, the central government, as well as professions within the state bureaucracies, have become the leading actors in the formulation and implementation of urban development plans.

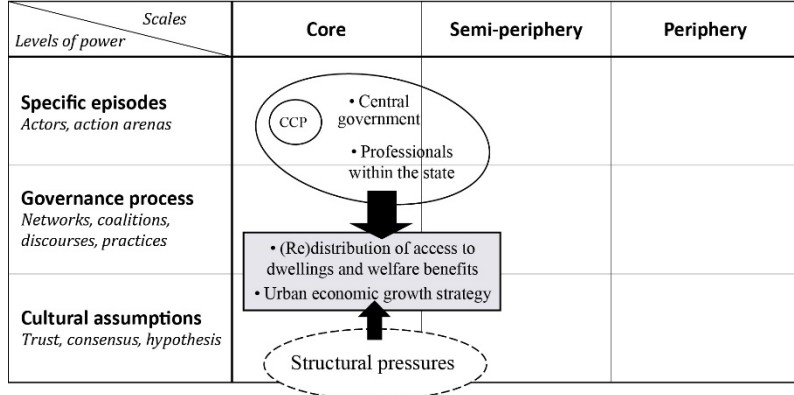

**Figure 5.** Transformation initiatives in governance dynamics (the 1980s): from party-directed economic growth to central-government-driven welfare products.

In this phase, the power structure of the authorities was reflected in the process of distribution of dwellings and welfare benefits. In this regard, the urban redevelopment policymaking in this period was also motivated by the structural pressures of urbanization

and economic growth. According to the principles of the Reform, the most important task for the authorities during this decade was to promote economic growth. The resulting policies, such as the "Dilapidated House Reconstruction", were mainly formulated to serve social welfare and the so-called public interest. At the local level, the authorities accordingly launched their own urban development policies to serve the economic priorities and ideological imprint at the central level, such as the land-transfer policy in edged areas from rural land-use to urban construction land-use, public-invested infrastructure construction, and so on. Meanwhile, the state-owned enterprises (SOEs) or "work units" ("*danwei*") were the main actors responsible for policy implementation. For instance, after the central government launched the housing system reform in June 1980, the Guangzhou Municipal Government soon released the policy "Six unified" (i.e., unified planning, unified land acquisition and demolition, unified design, unified construction, unified supporting-infrastructure, and unified management) in order to strengthen the role of the government in urban (re)development practices and promote the efficiency in the supply of dwellings. As a result, several large residential communities such as the "Huaqiao Community" and "Jiangnan Community" in the Yuexiu District were designed and constructed by an SOE enterprise—the Guangzhou Urban Construction and Development Corporation.

### 4.2. Property-Led Redevelopment Strategy (the 1990s): From a Welfare-State to a Public–Private Partnership

During this decade, the multi-level governments continued the growth-oriented urban (re)development policies. With the deepening of the privatization of housing and the reform of the tax system, land finance became a major financial source for local governments. Especially after Jiang Zemin proposed the idea of the "Three Represents" ("*Sange Daibiao*") in the late 1990s, a new change took place in China's political system, namely the transition from the proletarian dictatorship to meritocracy. In such a system, government measurement is based on its performance, which is measured through examinations or demonstrated achievement [51]. As a result, the hierarchal relationship between the central government and local governments took shape. To accumulate political capital within the inter-city competition, local government officers paid more attention to land development programs. Take Guangzhou as an example, whose resulting policies led to large-scale demolition and reconstruction in the inner-city areas. In the late 1990s, many inner-city neighborhoods in Liwan District were demolished and rebuilt as mixed commercial plazas or high-density residential communities, such as the Liwan Plaza and the Hengbao Huating community. The success of these projects mainly depended on the local-level land readjustment policies, in which local governments integrated previously fragmented land resources and transferred them to real estate developers for land value increments. As a result, the property-led redevelopment policies were institutionalized to support rapid urbanization and real estate development. In this phase, the decision-making power regarding urban (re)development strategy has been transferred from central government to a more vested interest group including local government and private sectors, as shown in Figure 6.

In the past two decades, urban scholars have deeply analyzed this property-led redevelopment policy in China in terms of its background, motivation, and consequences [52,53]. It is argued that the boundaries between the public and private sectors have been blurred and that the power of decision-making is vested in multiple professional interest groups, including private sectors. In this stage, the property-led demolition and redevelopment have given rise to much social controversy and resistance from local residents who own the houses. These homeowners, as well as other vested stakeholders, form the periphery of the governance network, as shown in Figure 6.

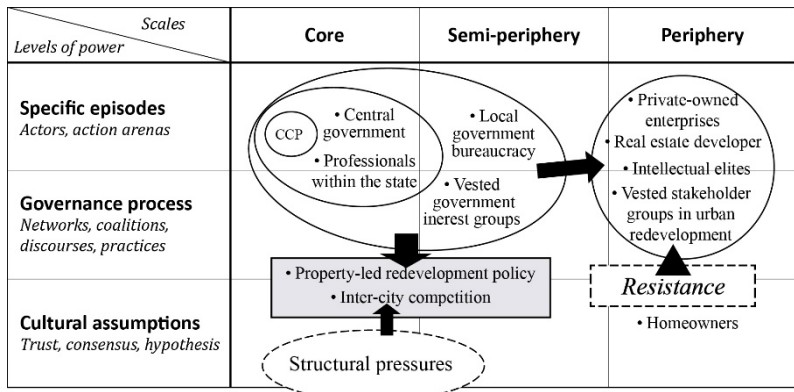

**Figure 6.** Transformation initiatives in governance dynamics (the 1990s): from a welfare-state to a public–private partnership.

Despite the significant contribution to urbanization and economic growth, these property-led redevelopment projects during this process resulted in serious damage to historical heritage, socio-spatial relations, and cultural environments. Moreover, in the late 1990s, due to the global economic downturn, the market-led (re)development failed to address the financial crisis and left many uncompleted projects in the demolished parts. In response, the Guangzhou Municipal Government rejected real estate developers from urban (re)development projects in the inner-city areas. Meanwhile, the municipal government released the "Implementation Plan of Guangzhou Dilapidated and Dangerous House Reconstruction" in 1999 to further highlight the role of the government in urban redevelopment practices in terms of planning, investment, and implementation. It was not until 2005 that market-based social capital re-entered the urban redevelopment arena.

*4.3. Area-Based Regeneration Strategy (since the Late 2000s): From Demolition-Based Renewal to Community-Based Petite Redevelopment*

In the late 2000s, the focus of Guangzhou's urban development policies transformed from incrementalism-based expansion to inner-city regeneration. During this process, this study suggests that there were three main phases given relevant redevelopment policies:

(1) From 2004 to 2009: In the six years of preparation after winning the bid in 2004, the authorities in Guangzhou strategically used the Asian games as a vehicle to fulfill its development goals. To further support the city's rebranding movement and inner-city regeneration, the Guangzhou Municipal Government proposed a new urban strategy at the 9th National Congress of the CCP in Guangzhou (December 2006)—the "Zhongtiao" ("revalorizing the city's central area"). According to official documents, there were about 1.038 million $m^2$ of "dangerous and dilapidated houses" in Guangzhou's old city area (involving three districts named Liwan, Yuexiu, and Haizhu). The redevelopment projects of these dilapidated buildings houses were the focus of the "Zhongtiao's" urban development strategy. Enninglu was one of the first pilot projects under this strategy.

(2) From 2009 to 2015: In December 2009, a large-scale urban redevelopment scheme named the "Three Old Renewals" ("*Sanjiu Gaizao*") was facilitated by the Guangdong Provincial Government in terms of a top-down land-use [33]. The resulting policies highlighted the area-based regeneration approaches, which means that the focus of the redevelopment project expanded from the specific material space to the mixed socio-economic space in a broader area. However, the area-based policy still required large-scale demolition. For instance, in the first demolition plan in 2007 in Enninglu, 82% (115,000 out of 140,000 $m^2$) of buildings that were planned to be demolished were not even identified as "dangerous or dilapidated houses". These demolition-based redevelopment policies were met with intense social resistance by local communities, historical conservationists, and other social interest groups, as shown in Figure 7.

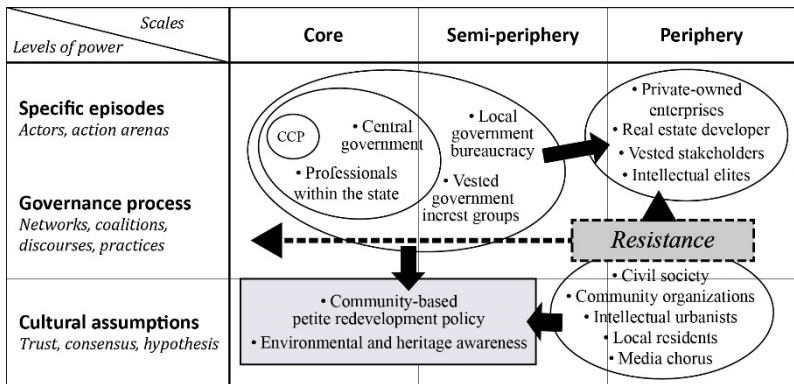

**Figure 7.** Transformation initiatives in governance dynamics (since the 2000s): the rise of civil society.

(3) Since 2015: In February 2015, the first urban redevelopment agency in China (the "Urban Renewal Authority of Guangzhou") was established to assume responsibility for the original agency of the "Three Old Renewals". The establishment of this government agency aimed to explore new forms of urban renewal and provide a platform to coordinate the diverse key actors. As an on-going project, the regeneration of Enninglu, therefore, became a priority for local authorities and local planners, and thus local politics have been dominated by public sector agencies that are intent on promoting particular forms of the middle class, consumption-based regeneration. In September 2015, the "Measures of Guangzhou Municipality on Urban Renewal", which was issued by the municipal government, proposed a new urban renewal approach of petite redevelopment. In contrast with the previous demolition-led approach of the "Three Old Renewals", the new pattern of petite redevelopment referred to scattered-site demolition, conservation/restoration of existing buildings, and the upgrade/replacement of spatial functions in order to focus more on the improvement of living conditions and the conservation of historic/cultural features. In this context, the Yongqing Lane area, as shown in Figure 2, was launched as the first pilot project of this new form of petite redevelopment.

Despite more than a decade of debate, the Enninglu redevelopment is still struggling for an adaptive redevelopment strategy. In April 2016, Vanke, one of the largest real estate developers in China, signed a 15-year rental contract with the Liwan District Government to redevelop a "cultural and creative quarter with commercial functions" in Yongqing Lane. By turning public housing into boutiques, studios, coffee shops, and B&B hotels to cater to the aesthetic tastes of the "creative class", the Yongqing Lane area has become a hot spot for young tourism and petite bourgeoisie consumers. As such, Yongqing Lane has a unique complexity. Under the distinct circumstances of a mixed community composed of original residents, tourists, and small-scale creative-oriented businesses, it reflects an emerging symbiotic pattern of contradiction among multiple stakeholders. On 16 August 2018, the second round of urban redevelopment projects in Enninglu, as shown in Figure 2, was launched by the Guangzhou Liwan District Government. In this stage, Vanke signed a 20-year lease contract with the Liwan District Government to develop a new Xiguan historical and cultural creative center on the demolished parts of Enninglu. With more than RMB 10.7 billion of investment, the redevelopment project plans to take advantage of the 70,000 m$^2$ of public housing and 6.6 ha of vacant land resources in Enninglu to develop a creative district with mixed-use functions that will comprise commercial, recreational, and historical/cultural education activities. The Urban Renewal Project Office of Liwan District (managed by the Urban Renewal Authority of Liwan District), which owns most of the site to the north of Enning Road (including public housing and vacant land) became a key player in facilitating the newly established redevelopment project.

Unlike many other finished projects in the past two decades, Enninglu has not yet been completed because of social resistance and insurgent practices during both the demolition

and reconstruction processes. Especially after the Real Property Law was released in 2007, the demolition-based urban renewal in urban China has been met with intense resistance. Many urban scholars have investigated the phenomenon of "nail houses" ("*dingzi hu*") to examine the injustices in China's redevelopment practices (for example, see [24]). Yet, the actions of the "nail-houses" are typically characterized by individual wills and can be addressed by economic compensation. According to the field investigation in the Enninglu redevelopment project, this study argues that there exists a networked relationship among key actors within the insurgent practices, as shown in Figure 7. In this case, participants on the periphery of the governance network are beginning to expand from individual homeowners (individual opposition) to a pluralistic social group with more civil society overtones (vested interest groups work with each other and form a discursive coalition). For instance, this study found that participants in civil society including residents, the local media, intellectual urbanists, and non-government organizations have formulated a discourse coalition and relational network. Within this network, they share a high level of trust and consensus, they keep in contact with each other, and they take collective actions against the government's interventions. In the following section, this study further elaborates on the two waves of insurgent practices in Enninglu to analyze the relational network among these diverse actors.

## 5. Two Waves of Insurgent Practices during the Redevelopment of Enninglu

### 5.1. Against the Government-Led Demolition (2007–2011): The "Media Chorus" and the Rise of Civil Society

The period between 2007 and 2011 marks the most chaotic phase of the redevelopment when intensive collective actions taken by the local informal coalition resulted in an overhaul of the regeneration plan. The coalition contained housing and community activists who were opposed to renewal at their estates alongside the local media, social groups, and critical urbanists and planners. By specific episodes of governance, the coalition adopted diverse forms of resistance including open letters, joint-signed letters, posters, and a "media chorus" of critical commentaries on the government's performance. Many local newspapers (e.g., the *New Express* and the *Nanfang Daily*) reported the social concerns on the demolition–conservation conflict and on the social inequalities caused by displacement. In this stage, through collaboration with other social groups, local residents were mainly arguing for more compensation or to avoid displacement. From 2008 to 2011, the first wave of insurgent activities consists of three stages.

First, in May 2008, more than 80 households of residents in Enninglu submitted a petition to the National People's Congress to oppose the illegal demolition and defend their rights [54]. In response to these social concerns, the local government decided to adjust the regulatory plan and reduce the demolition area. To preserve the historic blocks, the plan changed the main roads that pass through the core area of Enninglu in the 2007 regulatory plan, which would directly result in large-scale demolition if implemented. As a result, Qilou Street, the Yongqing community, and some valuable historical dwellings were eventually preserved. Nevertheless, almost half of the buildings in Enninglu were torn down during this period.

Second, in January 2010, 183 households launched a campaign to oppose the redevelopment plan. They argued that the new planned commercial center in Enninglu undermined the original Xiguan culture, and that the illegal demolition damaged stakeholders' interests [55]. Further, in April 2010, 220 residents submitted an open letter to the municipal government requesting a revision of the compensation scheme and the redevelopment plan. According to an interview with one of the remaining residents in Yongqing Street in July 2017, the rental fee of the public houses was as low as RMB 200–300 per month (about 25 m$^2$ including the mezzanine floor), while the price of nearby housing was RMB 15,000–30,000 per m$^2$. Monetary compensation was almost impossible to meet the cost of living near Enninglu, while the available resettlement site was too far away for residents to continue their original work and lifestyles. Therefore, due to the huge gap in living costs, the resident I interviewed did not sign the relocation agreement and

continues to live in rented public housing. In fact, until now, some remaining residents are still hesitant about the future of Enninglu redevelopment, including one resident who owns a bronze shop on Enning Road since 1979. When he was asked about the attitude towards urban regeneration planning in Enninglu, he held a pessimistic view:

> *"Enninglu redevelopment planning is meaningless. Because all the neighbors were moved out after demolition, with only a few left. And what we say is useless because the plan is government-led. It is the government officials who made all the decisions".* (Interviews with the residents who are living and working in Enning Road, July 2017).

Further, this phase saw a rise in civil society groups in the insurgent practices in the Enninglu redevelopment project. For instance, during 2010 and 2011, two self-organized groups were formed which consisted of college students, academics, and social activists who called themselves the Enninglu Folk Focus Group and the Academic Focus Group, respectively [49]. These two groups have played an important role in preventing the demolition of old buildings, as well as in promoting awareness of the importance of public participation in the planning processes [56]. As a result, the insurgent actions in the Enninglu project received increased attention from the public and the central authorities, including from the former mayor Qingliang Wan and the central CPPCC member Minggao Wu [57,58]. Under the pressure of social resistance and a media attack, the local government had to readjust the redevelopment plan to increase the compensation and provide more dwellings for in-situ relocations. In addition, this plan proposed a new urban regeneration policy of "self-help redevelopment by residents". In June 2011, 130 households signed an open letter to the mayor to support the self-help redevelopment model and made further recommendations on the implementation rules [59]. Although it did not receive the final approval from the municipal government, the public attention to this project entailed the success of the insurgent practices during this period. Given that the revised plan was designed by higher education institutions, this study argues that the higher education institutions and experts began to engage in this game as a "professional interest group".

*5.2. Against the Consumption-Oriented Renewal (2016–2018): A "Professional Interest Group" and the Emerging Network Society*

The recent struggles over the consumption-oriented strategy of abandoned dwellings' utilization in Enninglu have sparked a new wave of social dissent. The arena of specific governance episodes has expanded to a network society and relied on the development of social media. During this process, the insurgent practices have further forced the transformation in governance by arguing for public participation and enhancing the role of the "professional interest group" which contains planners with professional education backgrounds, experts, and design institutions. For instance, in the negotiation meetings, the representatives of local media thought highly of another redevelopment project just because the planner has an educational background in community-based development, and she particularly emphasized the role of experts in the policymaking process:

> *"Each version of the redevelopment planning and anything within this plan must be reviewed by a group of experts, including the architecture style, the dwellings that need to be demolitioned, the dwellings that need to be rebuilt, etc. And we will not endorse the program until you provide consent from the expert panel".* (Comments by a journalist in the committee meeting, 7 January 2019).

Since 2015, the Enninglu redevelopment project has been re-started by the Liwan District Government. This newly-established agenda consists of two phases, both of which have caused a set of insurgent actions by the local coalition but with different characteristics. First, in August 2015, the redevelopment project in Yongqing Lane was issued by Liwan District as a pioneer of petite redevelopment, as shown in Figure 2. Between 2015 and 2016, Vanke invested more than RMB 65 million to renovate 50 public houses—which were confiscated by local authorities in the previous demolition phase—which were turned into establishments that cater to the aesthetic taste of the "creative class". After regeneration, the rents in this block met a significant increment from 30–40 RMB/m$^2$ to 80–150 RMB/m$^2$,

while the rents in the nearby area also tripled due to gentrification. The commercial vitality gradually increased after a short plateau and reached its peak in 2018 after the state leader Jinping Xi visited Enninglu. However, due to the lack of public participation in the policymaking, the redevelopment project in Yongqing Lane has also led to intense social dissent. In February 2017, four representatives submitted a joint objection letter, signed by 60 households of Enninglu residents, to various government agencies including the municipal government, the district government, and the Guangzhou Municipal People's Congress Standing Committee. I interviewed a resident in one of these 60 households, who still lives in rented public housing in Yongqing block. She became emotional when she was asked about the influence of Yongqing Lane redevelopment project:

> *"The program has damaged my life here! The next-door has been turned into a coffee shop and it is so noisy! They (Vanke) know that, but they did nothing to avoid it (the noise). Moreover, no one told me before the redevelopment that there would be a commercial center here, they (Vanke) start the construction without any discussion with us. If it is not because of the low rental fee (of public social housing), or if it is possible to get a nearby resettlement house, I also would be willing to move away".* (Interview with one of 60 households in July 2017).

Second, in August 2018, a new stage of the urban redevelopment project in Enninglu started. In this stage, the Urban Renewal Authority of Liwan District, as the local planning authority, played a pivotal role in facilitating the public-engaged collaborative regeneration practices. On 7 September 2018, the "Enninglu Co-Creation Committee" was officially established by the Liwan District Government to promote the collaborative relationship between diverse actors. The Committee comprises 25 members and includes 15 resident-representatives, 1 government official, 1 developer administrator, 2 planning officers of Liwan District Planning Bureau, 3 planning experts, 1 community planner, and 2 representatives from the local newspaper. More importantly, the Committee has established an online discussion group on social media. In so doing, it provides a platform for stakeholders to express their demands directly to decision-makers, and for other members, including local media and experts, to oversee how the government officials have responded to and addressed residents' requests. As a local planner pointed out in an interview, "The Committee provides access for residents to contact decision-makers directly and express their demands. Collaborative planning aims to make this place better through cooperation, not quarrel" (personal communication, 21 May 2017, Enninglu). In this sense, a network society has emerged in this process.

However, some people still view the Committee as another political tool with which to legitimize the hegemonic policymaking, and they believe that the ongoing redevelopment projects are still outcome-oriented with the aim of having quick success and instant benefits. More importantly, the original demolition plan is still in effect, despite having been postponed since 2009. Given that housing transactions during the demolition period are frozen, the private housing in the "red line" area (demolition area) cannot be sold or reconstructed. As one of the resident-representatives expressed his anger and concerns in the meeting, who argues for equal rights in private housing reconstruction by residents to public housing reconstruction by the government:

> *"While the Canton Tower in the new city center was raising, this place is falling... Over the last 13 years, we (residents) continually expressed our demands and nobody cares. We don't need a vanity project for someone's political achievements. The original residents have suffered a lot due to the implementation of this national paradigm project. To some extent, the pilot policies launched by the central government have become a shield for local authorities. And I reckon that this committee, this so-called public participatory scheme, in part, is the cover of capital development actions. The 13-year redevelopment process of Enninglu has witnessed the utilitarianism of urban development. But it cannot happen again this time. We residents, as the owners of private housing, are literately equal to the owners of public housing. So why can they carry out new structures without construction*

*permits while we residents cannot".* (Comments by the resident representative in the meeting, 22 May 2019).

## 6. Discussion: Discourse Coalition and Governance Transformation

By elaborating on the two waves of insurgent practices during the Enninglu redevelopment process, this study argues that the emergence of community-based governance initiatives suggests an emerging shift towards substantive participatory governance in the Chinese context. In the first wave, the opposition coalition took shape during the insurgent actions against the government-led demolition. Despite some societal groups arguing for increased awareness of historical conservation, the residents were mainly fighting for compensation or avoiding displacement. Thus, they needed support from other civil society actors including the media, focus groups, and active intellectuals to obtain increased social attention by scaling-up as a public event. In the second wave, however, the local coalition tended to protect the current local scale, whereas the government tended to argue for the greater interest through scalar reconfiguration. In so doing, the local coalition conducted insurgent actions for deepening community empowerment and argued for equal rights in private housing reconstruction by residents to public housing reconstruction by the government. All in all, the regeneration of the urban spaces in Enninglu has been a result of a combination of local agenda-setting and market opportunities. The operationalization of regeneration agendas at the local level has involved the formation and re-formation of bilateral and multi-lateral networks, which have been brought together at specific times to facilitate regeneration. This has occurred for issues such as construction procedures and self-help reconstruction policies, which have been re-aligned to support broader inner-city regeneration strategies. While the local authority and developers have been the dominant actors, other interests—particularly the counter-discourse coalition which comprises local media, residents, and professional experts—have also played a major part in facilitating their own diverse interests.

In this process, three discourse-based approaches can be identified: (1) An official/mainstream discourse of the local government that dominates both policy approaches and urban place-based planning practices; (2) an oppositional/critical discourse that is prominent among housing and community activists who are opposed to renewal at their estates, as well as the local media and some academic critical urbanists; and (3) an emerging and supportive discourse driven by residents' pursuits of a better life and planners' visions of ideal community conditions. Meanwhile, during the redevelopment period, discourse-based coalitions were also formed. On one hand, the most representative was the unique but strong coalition formed in the social resistance process of Enninglu, which comprised of local media (mainly represented by the New Express) and residents who suffered from the demolition. This discourse-based coalition played an important role not only in the demolition stage, but also in the newly established redevelopment phase. In particular, the local media played a vital role in framing the social issues of demolition, planning, negotiation, and cultural conservation because they reflected the extensive opinions of residents through publications and transferred planning-related knowledge to help the residents express their demands. Most importantly, more and more people have paid attention to the Enninglu redevelopment project due to the residents' stories that were reported by the local media. Consequently, the continuous attention of the local media has expanded the social impacts of Enninglu and thereby has further promoted the discourse transformation of the Enninglu redevelopment project. On the other hand, agendas for the Enninglu regeneration project have been led by the local authority with developers and other interests playing a secondary, yet supportive role. In the context of the profit reconfiguration based on the policy-oriented profit supply, the local authorities, including the Liwan District Government and local urban planning agencies, are forming a dominant discourse coalition with real estate developers under the pursuit of profit in specific urban redevelopment projects. Some critical planners have even criticized that this discourse coalition has used the "Co-Creation Committee" as the concealment of its actual profit-

pursuing approaches in the new Enninglu redevelopment project. This kind of comment makes sense, since the redevelopment plan issued by Wood and Zapata, Inc., reflects the current urban development perception of the local authorities. As the only two actors that can intervene with spatial change directly, the local planning authorities and developers are in close contact with the planners and construction/operation. In March 2019, a new planning agency—the Guangzhou Bureau of Planning and Natural Resources—was established in the state-level institutional reform. Meanwhile, the Guangzhou Urban Renewal Authority was revoked and merged with this newly established planning authority. In this case, the urban redevelopment in Enninglu is facing a strategic transformation based on the emerging discourse of sustainable redevelopment in a more socially responsible way.

## 7. Conclusions

The case study of Guangzhou shows that the mechanism of urban redevelopment policymaking in China has experienced a profound shift. The governance actors involved have gradually evolved from a government-centered approach to a multi-scalar, pluralistic governance network. In this context, the specific episodes contributing to governance transformation refer to both formal and informal initiatives. Over the past three decades, government-backed urban renewals in China have been regarded as a powerful instrument to address the shortages of urban land resources and dwelling supply. During this period, the government acted as a "welfare state" to supply dwellings and serve the public interest. Several great cities in China, including Guangzhou, took shape as a result of being driven by this kind of property-led redevelopment strategy. More recently, many insurgent actions have occurred in urban China against the government-intervened projects, such as demolition-led real estate development, refuse dumps during infrastructure construction, and all kinds of other urban agendas that occur without comprehensive public participation. Under these circumstances, the contemporary urban redevelopment policies in China have moved from a focus on "Dilapidated House Reconstruction" ("*Weifang Gaizao*") and property-led urban renewal to citizen empowerment and area-based regeneration projects. These changes manifest themselves in two ways. First, the planning regime has transited from state-dominant practices to one which is primarily driven by the local government, the enhanced role of higher education institutions and experts as a "professional interest group", and the increased participation of non-state actors in the policymaking process. Second, the decision-making mechanism has transformed from being interventionism-oriented to a polyarchy-oriented system in which both the advocacy coalition and opposition coalition are embedded in the governance network. In contemporary China, urban regeneration involves more intricate actors due to the privatization of property-rights during globalization and neoliberalization. In particular, regeneration practices play a key role in reinvigorating inner-city areas through the re-population of urban spaces and the attraction of new forms of investment. However, there exists a mismatch between the path dependency of the inherited structures and the growing awareness of citizen empowerment. To address the existing gaps in the regulatory sphere, the role of professional experts has been intensely highlighted not only by the governments, but also by the grass-root groups in society. On one hand, the government agencies rely on experts to make the blueprint, to legitimate the planning scheme, and to operate with other actors. On the other, the grass-root groups that act as regulatory bodies also rely on intellectuals' participation to legitimize their insurgent practices.

Two waves of insurgent practices in Enninglu in Guangzhou further reveal an emerging relational social network in contemporary urban regeneration. This case study adds a new dimension to the network governance theory by highlighting the process of insurgent activities in local redevelopment practice and of building discursive coalitions among different actors in an authoritarian context. Unlike western democratic countries, dissident activism for city rights in China has experienced a difficult process of development, shifting from individual resistance to networked activism. Only in recent years has there been a tendency in China's redevelopment practices for community activists and

residents to join forces with each other to generate a relational network against inherent structures. However, in the Chinese context, the role of professionals is emphasized by both the advocacy coalition and the opposition coalition. On the one hand, this study argues that "professional interest groups" have brought together planning experts, practitioners, scholars, college students, and other societal groups to foster the transformation in terms of discourse, policy, and governance structure. On the other hand, two discursive coalitions also regarded the participation of professionals as a basis of legitimacy to defend their positions. As such, the sustentation and legitimization of both specific episodes of governance and the governance process can contribute to cultural assumptions about appropriate agendas and practices held by different civil society actors. It is through these assumptions, as recognized by civil society actors themselves and the "media chorus" of critical commentary on government performance, that the core of insurgent practices are constituted. In view of the insurgent practices in the Enninglu redevelopment process, the civil society actors indicate the accumulation of cultural assumptions by challenging the legitimacy of government actions and holding them to account in terms of both formal (participation, regular meeting, representative selection, etc.) and informal procedures (resistance, joint petition, media coverage, etc.).

To conclude, as empirical research that has been continuously tracked for three years, this paper adds to the network governance literature by highlighting both the formal and informal scenarios in China's governance transformation and providing evidence for the possibility of constructing a more open democracy in the planning regime. Yet, this research has limitations in two aspects. First, the arguments of this research are based on a single case study and therefore may be contingent or idiosyncratic. Second, this research adopts the PAR approach to conduct the field investigation. In this process, the researchers are deeply engaged in the planning making process and communicated directly with those involved in the insurgent activities. Thus, the data we collected in this process is, to some extent, subjective and emotional due to the individual bias of the residents. These shortcomings are expected to be added to and improved by ever-expanding data collection and more case studies and investigations.

**Author Contributions:** Conceptualization, N.Z. and Y.L.; methodology, N.Z. and Y.L.; investigation, N.Z.; data curation, N.Z.; writing—original draft preparation, N.Z.; writing—review and editing, Y.L. and J.W.; supervision, Y.L. and J.W. All authors have read and agreed to the published version of the manuscript.

**Funding:** This research was funded by the National Natural Science Foundation (Grant No. 52008171), National Natural Science Foundation (Grant No. 41771175), China Postdoctoral Science Foundation (Grant No. 2019M662914), Natural Science Foundation of Guangdong Province (Grant No. 2019A1515010717), Philosophy and Social Science Program of Guangzhou (Grant No. 2019GZGJ07), Guangzhou Think Tank for Guangdong-Hong Kong-Macau Greater Bay Area (Grant No. 2019GZWTQN03), National Key Research and Development Program (Grant No. 2018YFC0704603).

**Institutional Review Board Statement:** Not applicable.

**Informed Consent Statement:** Not applicable.

**Data Availability Statement:** Not applicable.

**Conflicts of Interest:** The authors declare no conflict of interest.

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
