# Peer review of "Network Governance and the Evolving Urban Regeneration Policymaking in China: A Case Study of Insurgent Practices in Enninglu Redevelopment Project"

_sustainability, doi:10.3390/su13042280_

Round 1

Reviewer 1 Report

The reference section is not always in alphabetical order. Authors should review the order of references.

I would advise authors to separate the "Discussion" section from the "Conclusions" section. In my opinion, a section with more explicit conclusions is missing.

Most of the figures are too blurred and some figures have some graphic noise. Authors should improve the graphic aspect of the figures.

Author Response

Many thanks for your kind reminder and your helpful comments! 

According to the three comments, I have revised the article as follows:

  • I have reviewed the order of the references and make sure they correspond to the text.
  • I redraw Figure 2 and Figure 3 to improve the graphic aspect of these figures and make them easier to read (line 239-244). For instance, I enlarged the text on the figures and simplified the digital information on the figures. In addition, I remove the original Figure 8 to make sure it would not confuse the readers. Instead, I explain it in plain English in the main text.
  • According to the reviewer’s suggestion, I separate the “discussion” section from the “conclusion” section to make it easier to understand (line 588-695). In the “discussion” section, I provide an extended discussion based on the findings of the Enninglu case study. In the “conclusion” section, I summarized the results and contribution of this paper in terms of both practical and theoretical aspects.

Reviewer 2 Report

The article has a promising introduction and reads very well. The story is presented clearly and in detail, but such empirical richness is not followed by strong theoretical elaboration. The section 6 on 'Discussion and conclusion' is too weak compared to the empirical material that has been presented (perhaps too extensively?) in section 4 and 5. In other words, the paper is too descriptive; it lacks abstraction and stronger links to the literature on governance and networks.

The clarity of the empirical part is also not mirrored in the figures that are quite confusing to me.

A final remark (and ethical concern) is about the way interviewees are referenced: I feel it does not assure much anonymity and confidentiality, particularly in a country with many specificities like China. Has this point been taken into account? Interviewees are referred to by their surname – or are those fictitious ones?

Author Response

Many thanks for your consideration and helpful comments! I appreciate that you enjoyed the introduction section. 

According to the three comments, I have revised the article as follows:

  • For the interview references, I thank the reviewer for his/her kind reminder. Thus I removed the surnames of those interviewees to protect their privacy (see line 474-483, 515-523, 540-549, 573-587). 
  • For the “discussion and conclusion” section, I thank the reviewer for his/her helpful suggestion. On this basis, I separate the “discussion” section from the “conclusion” section to provide more specific conclusions and statements. In the "conclusion" section, I attempt to better link the empirical results to the theoretical contributions of the network governance literature and summarize them in an abstract manner (see line 650-714). In the last paragraph, I added a discussion about the limitation of this paper and the qualitative methods (see line 650-714). In addition, to better links the empirical evidence with network theory, I added more explanation of Figure 6 and Figure 7 (see line 324-327, 430-436).
  • For the clarity of the figure in the empirical part, I thank the reviewer for his/her reminder. Therefore, I redraw Figure 2 and Figure 3 to make them easier to read. And I have removed the original Figure 8 to reduce the possible confusion. Instead, I have explained it in plain English in the “discussion” section.

Reviewer 3 Report

The paper “Network Governance and the Evolving Urban Regeneration Policymaking in China: A Case Study of Insurgent Practices in Enninglu Redevelopment Project” presents an interesting focus on the networked relations among involved actors in policymaking processes and on emerging insurgent practises in the Chinese context. Moreover, the paper clearly adheres to the Journal’s aims and scopes.

The article is laid out with sufficient clearness. All the key elements (abstract, introduction, data&methodology, results, discussion and conclusions) are present.

From a personal overview after the reading of the article:

-I do not see anything that needs essential re-thinking but I would recommend to consider some revisions. The paper is logical and the narrative smoothly flows from the beginning to the end.

-The applied methodology is quite elementary but appropriate if related to the stated objectives. However, potential limitations related to the methodology and datasets are not discussed.

Detailed comments

Just some comments that may help in improving the ms follow below:

-The references used in the introduction could include also more recent studies. Only six papers out of twentyfive have been published in the last ten years (The same goes for par.2).

-Please provide a good resolution images for Figures because they are really difficult to read and may result unclear.Moreover, all figures need to be described clearly and in more details, especially Figure 8. In fact, the first impression is that the text and the figures are not well related.

-Section 3 seems quite unbalanced compared to the preceding and the following ones. Actually, to properly assess the coherence of findings and thoroughness of discussion, more efforts should be put in highlighting the direct links between the applied methodology and obtained results. I would also suggest adding details about interviews, participatory focus group approach, observational works.

Author Response

We thank the reviewer for his/her detailed comments and his/her kind reminder.

According to the three comments, I have revised the article as follows:

  • I have added six pieces of literature published within 10 years in the “introduction” section (see line 53–80) and the literature review section.
  • For the “methodology and data collection” section, first, I added more details in the “data collection” section about the interviews, focus group experiences, and observational works that I adopted in this research (see line 254-268). Second, I added the limitation of the methodology in this research (see line 711-713). I also add some photos during these field investigations.
  • For the figure problem, first, I redraw all the figures to make them easier to read. Second, to enhance the links between texts and figures, I added more explanation for each figure (see line 284-285, 322-325, 432). Besides, to avoid confusion, I deleted the last figure (the original Figure 8). Instead, I explain it clearer in the “discussion” section (see line 614-651). 

Round 2

Reviewer 2 Report

The revised version has taken into account my previous remarks.

I confirm the strength of the article that has been sufficiently improved in the current version.

Author Response

We appreciate the reviewer's attention and all his/her helpful suggestions!  The comments help this article to build a better conclusion.

Reviewer 3 Report

Authors answers are fine with regard to the first point and part of the second point whilst it is opinion of the reviewer that all figures, although now clearly readable, are still not enough described and integrated into the text.

Moreover, references to limitations of the applied method (711-713) need to be rephrased because they are quite meaningless at the moment.

Author Response

We thank the reviewer for his/her contributed suggestions. According to the two comments, I modified the article in two aspects:

1. For each figure, I added more explanation in the text and try to make it match better with those elements in the diagram. Specifically, for Figure 5, see Line 278-299, 285-291. For Figure 6, see Line 317-335, 340-343. For Figure 7, see Line 432-438. I hope these texts can better explain the figures.

2. For the limitations in the conclusion section, I rephrase and explain more about the shortcomings and limitations of the methods applied in this research, including ethical risks, potential problems, and possible additions to future studies (see Line 719-730).

Again, we appreciate the reviewer's detailed comments, which helped a lot to improve the paper.